# Cell–cell interactions during the formation of primordial follicles in humans

Sylwia M Czukiewska[1], Xueying Fan[1] , Adriaan A Mulder[2] , Talia Van Der Helm[1], Sanne Hillenius[1],
Lotte Van Der Meeren[3,4], Roberto Matorras[5,6,7,8] , Cristina Eguizabal[9,10], Lei Lei[11], Roman I Koning[2] ,
Susana M Chuva De Sousa Lopes[1,12]

Gametogenesis is a complex and sex-specific multistep process during which the gonadal somatic niche plays an essential regulatory role. One of the most crucial steps during human female gametogenesis is the formation of primordial follicles, the functional unit of the ovary that constitutes the pool of follicles available at birth during the entire reproductive life. However, the relation between human fetal germ cells (hFGCs) and gonadal somatic cells during the formation of the primordial follicles remains largely unexplored. We have discovered that hFGCs can form multinucleated syncytia, some connected via interconnecting intercellular bridges, and that not all nuclei in hFGC–syncytia were synchronous regarding meiotic stage. As hFGCs progressed in development, pre-granulosa cells formed protrusions that seemed to progressively constrict individual hFGCs, perhaps contributing to separate them from the multinucleated syncytia. Our findings highlighted the cell–cell interaction and molecular dynamics between hFGCs and (pre)granulosa cells during the formation of primordial follicles in humans. Knowledge on how the pool of primordial follicle is formed is important to understand human infertility.

## Introduction

Oocytes and sperm cells are a group of cells specialized in passing on (epi)genetic information to future generations. In mice, upon arrival in the gonadal ridge, female fetal germ cells (FGCs) develop into cysts (or syncytia). There, sister FGCs stay connected through intercellular bridges, which mediate the exchange of organelles and cytoplasm during mitosis and meiosis, ensuring not only synchronization, but also playing a role in oocyte selection (Lei & Spradling, 2016; Ikami et al, 2021). During syncytia development in mice, only one or two FGCs per syncytium will be selected to form PFs, whereas the others serve as nurse FGCs, transferring the cytoplasmic content to the selected FGCs (Niu & Spradling, 2022).

In mice, the cysts are maintained until the FGCs reach the diplotene stage of meiotic prophase I (Pepling & Spradling, 2001). Thereafter, the cysts undergo fragmentation, to allow the formation of primordial follicles (PFs), that consist of a single oocyte surrounded by a layer of flat granulosa cells (Tingen et al, 2009). Interestingly, germ cells connected via syncytia or fragmented membranes in the absence of intercellular bridges observed in *Tex14* knockout female mice did not result in infertility, suggesting that intercellular bridges do not play a pivotal role in germ-cell connectivity and the generation of the oocyte pool or in oocyte quality and functionality (Greenbaum et al, 2009; Ikami et al, 2021).

In addition to intrinsic mechanisms of FGC development, such as down-regulation of *Sycp1* (Paredes et al, 2005), the process of PF formation also relies on the direct crosstalk between FGCs and (pre) granulosa cells. One of the known signaling pathways involved in the process of PF formation in mice is NOTCH signaling (Trombly et al, 2009). The conditional disruption of *Notch2* receptor (in the granulosa cell lineage) and *Jag1* ligand (in FGCs) in mice has previously been associated with the formation of follicles with multiple oocytes (Xu & Gridley, 2013; Vanorny et al, 2014), suggesting that NOTCH signaling is directly involved in the transition to PF in mice. Moreover, in mice and cynomolgus monkeys, it has been shown that gonadal somatic progenitor cells exhibit an epithelial–mesenchymal hybrid state upon specification (Sasaki et al, 2021). This hybrid state is caused by the activation of TGFB- and HIPPO-signaling pathways that contribute to epithelial-to-mesenchymal transition (EMT) while maintaining the expression of epithelial genes (Chang et al, 2019; Sasaki et al, 2021). EMT enhances the capacity of cells to migrate

---

[1]Department of Anatomy and Embryology, Leiden University Medical Center, Leiden, Netherlands  [2]Electron Microscopy Facility, Department of Cell and Chemical Biology, Leiden University Medical Center, Leiden, Netherlands  [3]Department of Pathology, Leiden University Medical Center, Leiden, Netherlands  [4]Department of Pathology, Erasmus Medical Center, Rotterdam, Netherlands  [5]IVIRMA, IVI Bilbao, Bilbao, Spain  [6]Human Reproduction Unit, Cruces University Hospital, Bilbao, Spain  [7]Department of Obstetrics and Gynecology, Basque Country University, Bilbao, Spain  [8]Biocruces Bizkaia Health Research Institute, Bilbao, Spain  [9]Cell Therapy, Stem Cells and Tissues Group, Basque Centre for Blood Transfusion and Human Tissues, Galdakao, Spain  [10]Biocruces Bizkaia Health Research Institute, Barakaldo, Spain  [11]Department of Obstetrics, Gynecology and Women's Health, University of Missouri School of Medicine, Columbia, MO, USA  [12]Department for Reproductive Medicine, Ghent University Hospital, Ghent, Belgium

Correspondence: lopes@lumc.nl

and invade by promoting the remodeling of ECM (Kalluri & Weinberg, 2009), whereas local changes in cell–cell adhesion drive tissue organization and cell sorting (Fierro-Gonzalez et al, 2013; Wickstrom & Niessen, 2018).

In contrast to mice, the development of female human fetal germ cells (hFGCs) is strongly asynchronous (Kurilo, 1981; Konishi et al, 1986). Late during the second trimester of pregnancy, the female gonads contain hFGCs in several developmental states with distinct molecular signatures in different gonadal zones or compartments (Li et al, 2017; Gomes Fernandes et al, 2018; Garcia-Alonso et al, 2022). Mitotic hFGCs (NANOG+, PDPN+, ALPL+, and POU5F1+) reside preferentially under the surface epithelium (outer cortex) of the developing ovary, whereas retinoic acid (RA)-responsive (DDX4+ and STRA8+) premeiotic hFGCs and meiotic (DDX4+ and SYCP3+) hFGCs (together late hFGCs) locate inside ovarian cords, that are structures lined by a COLIV+ basement membrane and surrounded by stromal cells, structurally resembling seminiferous tubes in males (Anderson et al, 2007; Heeren et al, 2015). After 16 weeks post-fertilization (WPF), PFs containing a single DDX4+ KIT+ TP63+ oocyte surrounded by granulosa cells start to emerge in the inner cortex (Fan et al, 2021).

Although a combination of single-cell and spatial transcriptomics of human gonads from the first and second trimesters has recently become available (Garcia-Alonso et al, 2022), we still have limited knowledge on how the cell–cell interactions between hFGCs and the gonadal somatic niche result in their transition from the ovarian cord into bona fide PFs. In this study, we investigated the cell–cell interactions between hFGCs and pre-granulosa cells during the transition from the ovarian cords into PFs. We discovered that hFGCs develop as mononucleated and multinucleated cysts, some being connected via intercellular bridges. Furthermore, as in mice, we observed epithelial–mesenchymal features in human pre-granulosa cells and identified genes that can distinguish pre-granulosa cells and bona fide granulosa cells. Finally, we identified changes in cell–cell adhesion and cadherin expression dynamics during the transition from ovarian cords to PFs in humans.

## Results

### Some hFGCs formed syncytia containing several nuclei without intercellular bridges

First, we determined the heterogeneity of hFGC, using keygenes POU5F1 (mitotic FGCs), DDX4 (post-mitotic FGCs), and SYCP3 (prophase I FGCs), and showed that it is resolved during the third trimester and that only PFs were present in the ovary after birth (Fig S1A and B). Next, using transmission electron microscopy (TEM), we showed that hFGCs in different stages of prophase I were confined to ovarian cords at 17–18 WPF (Figs 1A and B and S2A–D). Somatic pre-granulosa cells were positioned both at the edge of the ovarian cords in contact with the basement membrane, but also intercalated between meiotic hFGCs, whereas (fibroblast-like) stromal cells were located outside of the cords (Fig 1A). Surprisingly, we observed hFGC–syncytia containing several nuclei (2–3 nuclei)

without intercellular bridges both outside ovarian cords (interphase hFGCs) and inside ovarian cords (meiotic hFGCs) (Fig 1B). This was also observed in mice at embryonic days E14.5 and E17.5 where mFGC–syncytia without connecting intercellular bridges were also detected (Fig S3A and B).

Using immunofluorescence, we confirmed the presence of DDX4+CTNNB1+ and DDX4+pERM+ (pERM is phosphorylated ezrin–radixin–moesin) hFGC–syncytia without intercellular bridges inside the ovarian cords (Fig 1C) and multinucleated hFGC–syncytia without intercellular bridges consisting of ALPL+PDPN+ mitotic hFGCs in interphase outside ovarian cords at 17–18 WPF (Fig 1D). We quantified the number of mono and multinucleated DDX4+ hFGCs in 14–17 WPF (N = 4) and 18–20 WPF (N = 4) ovaries on paraffin sections and observed that about 7% of DDX4+ FGCs were present in multinucleated syncytia (Fig 1E), although this may be an underestimation, because of the fact that we are counting on sections. Using whole-mount immuno-staining for DDX4 together with phalloidin, we visualized the 3D structure of several hFGC–syncytia without intercellular bridges (Video 1). Interestingly, hFGC–syncytia without intercellular bridges contained either synchronized or unsynchronized hFGC nuclei (Fig 1F) and this was still the case in oocytes in PFs at 2 yr of age (Fig 1F).

### Some hFGC–syncytia showed intercellular bridges and contained apoptotic hFGCs

In addition to hFGCs forming syncytial structures without intercellular bridges, we also observed hFGCs connected by intercellular bridges, sufficiently large to allow mitochondria to pass through (Fig 2A). In TEM images, the diameter of the observed intercellular bridges ranged between 0.3–3.5 $\mu m$ (N = 3) (Fig 2A and B). Next, we investigated the localization of mitochondria (marked by TOM20) and Golgi complex (marked by GM130) in hFGCs between 16–20 WPF (Fig 2C). The (GM130+) Golgi complex did not display increased enrichment in specific hFGCs in ovarian cords and PFs; however, (TOM20+) mitochondria showed increased expression and accumulation in hFGCs during the transition between ovarian cords and PFs (Fig 2C).

To further characterize the intercellular bridges, we used KIF23 (also known as MKLP1), a marker of intercellular bridges in mouse and human male adult spermatocytes–spermatids (Greenbaum et al, 2007) (Fig 2D) and confirmed the presence of intercellular bridges between DDX4+ hFGCs in the ovarian cords of fetal ovaries (Fig 2D). In addition to intercellular bridges, KIF23 also marks midbody remnants (Halcrow et al, 2022); therefore, we evaluated the expression of *TEX14*, an additional marker of stable intercellular bridges in germ cells (Greenbaum et al, 2007). To this end, an online available single-cell transcriptomics dataset was used (Li et al, 2017) and 617 cells corresponding to 18–26 WPF fetal ovaries were extracted. After clustering, the cells were visualized using uniform manifold approximation and projections (UMAP) and the three main cell types as previously reported were identified (Li et al, 2017; Fan et al, 2021): *WT1+COL3A1+* gonadal somatic cells, *CD68+CD4+* immune cells, and hFGCs in different developmental stages (Fig S3C and D). As expected, *TEX14* was not expressed in the gonadal somatic and immune cells, but was highly expressed in meiotic hFGCs (Fig 2E). In agreement,

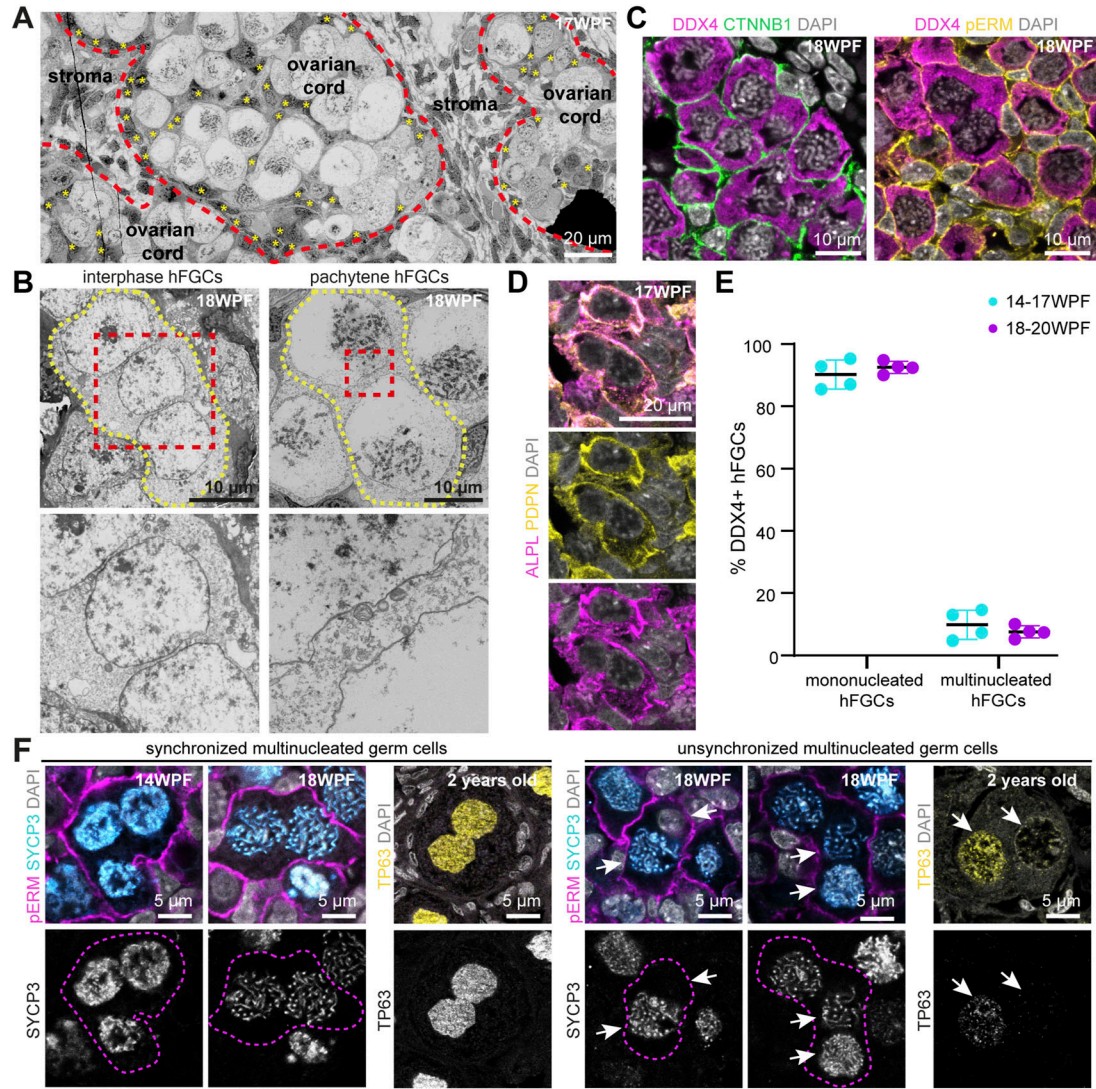

**Figure 1.  Human fetal germ cell (hFGC)–syncytia in human ovarian cords.**
**(A)** Representative transmission electron microscopy image of human ovarian cords. Cord boundaries between ovarian cords and stroma are outlined with dotted red lines. Pre-granulosa cells inside ovarian cords are marked with yellow asterisks. **(B)** Transmission electron microscopy images of hFGCsyncytia without intercellular bridges showing nuclei at interphase and pachytene. Cell membranes are highlighted with yellow dotted lines. Dashed box is shown magnified below.
**(C)** Immunofluorescence for DDX4/CTNNB1 and DDX4/phosphorylated ezrinradixinmoesin (pERM) in 18 WPF ovary. CTNNB1 and pERM label the cell membrane; DDX4 label hFGCs in the ovarian cords. **(D)** Immunofluorescence for PDPN/ALPL in 17 WPF ovary. PDPN and ALPL label the cell membrane of mitotic hFGCs. **(E)** Percentage of mononucleated and multinucleated DDX4+ hFGCs in 14–17 and 18–20 WPF ovaries (mean ± SD). **(F)** Immunofluorescence for pERM/SYCP3 or TP63 in ovaries of different ages showing synchronized (left panel) and asynchronized (right panel) multinucleated germ cells. The single channel for SYCP3 or TP63 is shown at the bottom. SYCP3 marks meiotic hFGCs and TP63 marks oocytes.

expression of *TEX14* was also confirmed by RNA fluorescent in situ hybridization (FISH) in pERM+ hFGCs (Fig 2F).

Most of hFGC–syncytia observed at 17–18 WPF did not display signs of apoptosis regarding nucleus or mitochondria morphology in TEM images (Fig 2G) and most meiotic DDX4+ hFGCs–syncytia were negative for cleaved CASP3 (cCASP3) in contrast to neighboring apoptotic cells (Fig 2H). After quantification on paraffin sections, we showed that at 14–17 and 18–20 WPF about 4% of the total number of DDX4+ FGCs were cCASP3+, with the majority corresponding to mononucleated hFGCs (Fig 2I). Although this may be an underrepresentation, this percentage of apoptosis is comparable

with what has been reported in mice, where a different type of programmed cell death (by acidification) was reported (Niu & Spradling, 2022).

## Transition from pre-granulosa cells in ovarian cords to granulosa cells in PFs

When examining the association between hFGCs and pre-granulosa cells inside the ovarian cords in TEM images, we observed that pre-granulosa cells showed thin membrane protrusions that engulfed

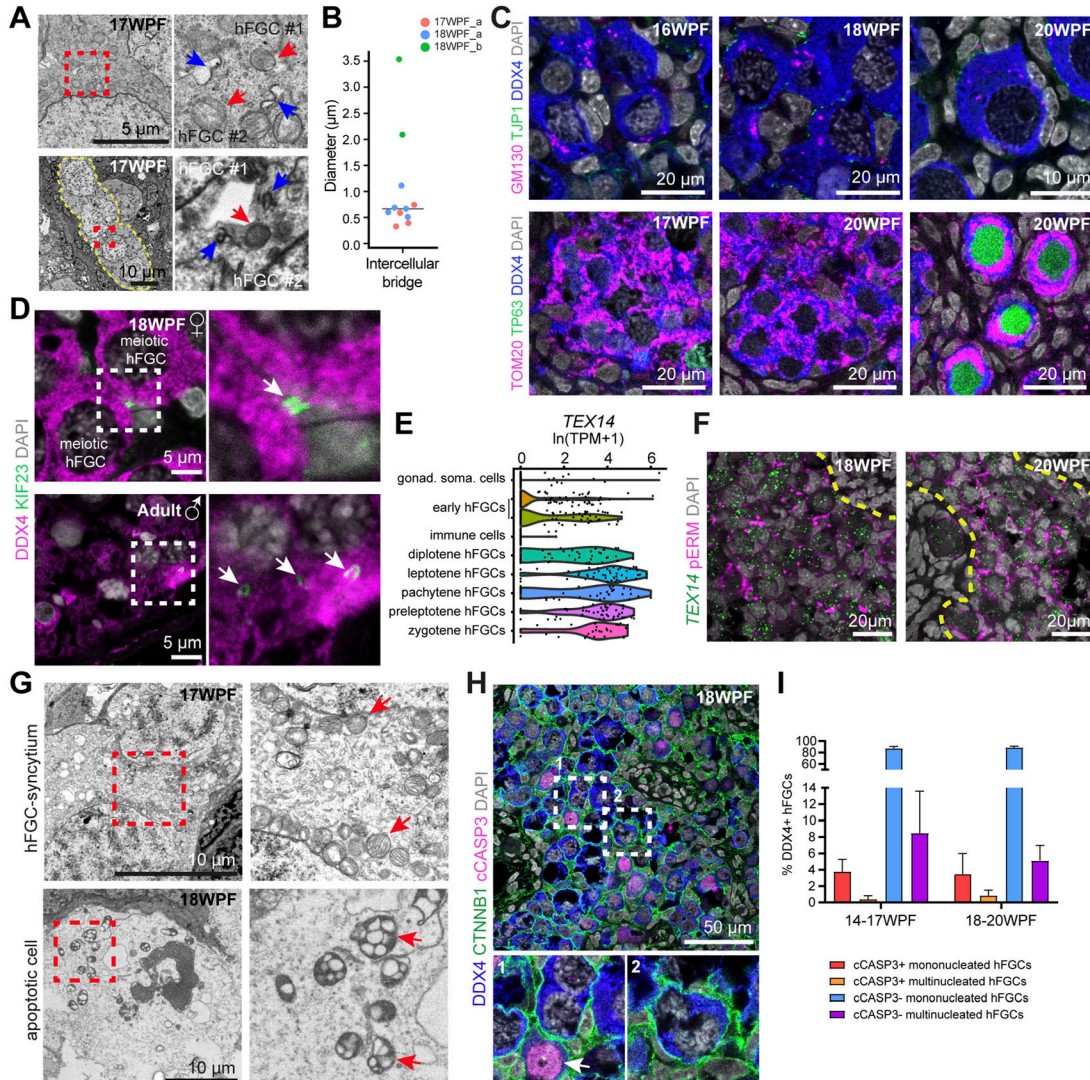

**Figure 2. Human fetal germ cell (hFGC)–syncytia show intercellular bridges and constriction by pre-granulosa cells.**
**(A)** Transmission electron microscopy (TEM) images of intercellular bridges between adjacent hFGCs (top) and hFGCsyncytium (bottom). Dashed boxes are shown magnified to the right. Red arrows point to the mitochondria. Blue arrows point to the intercellular bridge between adjacent hFGCs. **(B)** Quantification of the diameter of intercellular bridges in TEM images of 17 WPF_a, 18 WPF_a, 18 WPF_b ovaries. Median is shown as the black line. **(C)** Immunofluorescence for GM130/TJP1/DDX4 in 16, 18, and 20 WPF ovaries (top) and immunofluorescence for TOM20/TP63/DDX4 in 14, 17, and 20 WPF ovaries (bottom). GM130 marks the Golgi complex, TOM20 marks the mitochondria, and DDX4 marks hFGCs. **(D)** Immunofluorescence for DDX4/KIF23 in 18 WPF ovary (top panels) and adult testis (bottom panels). Dashed boxes are shown magnified to the right. Arrow points to KIF23+ DDX4+ germ cells. **(E)** Violin plot show the expression (in ln of transcripts per million + 1) of *TEX14* in different germline and soma clusters. Each dot is a single cell. **(F)** RNA FISH for *TEX14* combined with immunofluorescence for phosphorylated ezrinradixinmoesin in 18 and 20 WPF ovaries. Yellow dashed lines mark the border of ovarian cords. **(G)** TEM images displaying normal morphology of multinucleated hFGCs (top panels) and apoptotic cells (bottom panels). Dashed box is shown magnified to the right. Arrows point to the mitochondria. **(H)** Immunofluorescence for DDX4/CTNNB1/cCASP3 in an 18 WPF ovary. Dashed boxes are shown magnified at the bottom. Arrow points to cCASP3+ DDX4+ hFGC. cCASP3 marks apoptotic cells. **(I)** Percentage of mononucleated and multinucleated DDX4+ hFGCs in 14–17 and 18–20 WPF ovaries (mean ± SD) in apoptosis (cCASP3+) (mean + SD).

individual hFGCs, impeding immediate contact between adjacent hFGCs (Fig 3A). Notably, we discovered that pre-granulosa cells showed cell-membrane protrusions that were constricting into the meiotic hFGC–syncytia without intercellular bridges (Fig 3B). These constrictions were also observed in mice by TEM (Fig S3B).

Various forms of membrane protrusions are characteristic of cells undergoing EMT (Mattila & Lappalainen, 2008; Murphy & Courtneidge, 2011). In mice and cynomolgus monkeys, it has been shown that gonadal progenitor cells exhibit an epithelial–

mesenchymal hybrid state (Sasaki et al, 2021). In humans, this was also the case as FOXL2+ pre-granulosa cells, both inside and outside ovarian cords at 12 and 16 WPF, co-expressed mesenchymal marker VIM and epithelial marker CDH1 (Fig 3C), indicating both epithelial and mesenchymal properties. To further confirm the epithelial–mesenchymal hybrid state of human pre-granulosa cells, we investigated the HIPPO signaling pathway activation, known to induce EMT (Chang et al, 2019) and the expression of panKRT, a marker of epithelial cells (Menz et al, 2022). We observed

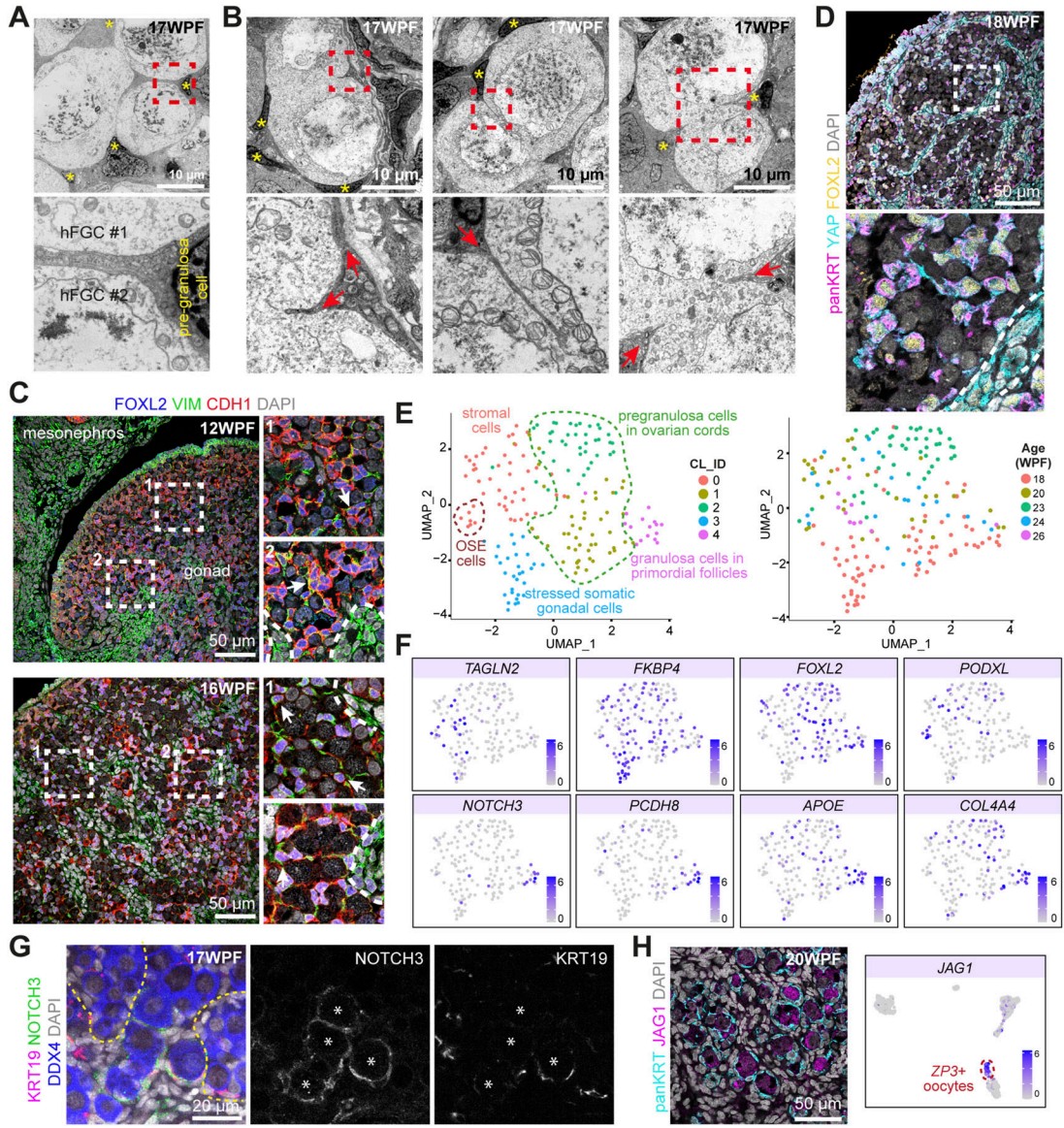

**Figure 3. Characterization of human pre-granulosa and granulosa cells subpopulations.**
**(A)** Transmission electron microscopy images of pre-granulosa cells intercalated between meiotic human fetal germ cells (hFGCs) inside ovarian cords in a 17 WPF ovary. Pre-granulosa cells inside ovarian cords are marked with yellow asterisks. Dashed box is shown magnified at the bottom. **(B)** Transmission electron microscopy images of pre-granulosa cells constricting meiotic multinucleated hFGCs inside ovarian cords in a 17 WPF ovary. Pre-granulosa cells inside ovarian cords are marked with yellow asterisks. Dashed box is shown magnified at the bottom. Red arrows indicate protrusions of the pre-granulosa cells. **(C)** Immunofluorescence for FOXL2/VIM/CDH1 in 12 WPF (top panels) and 16 WPF ovaries (bottom panels). Dashed boxes are shown magnified to the right. Arrow points to triple-positive pre-granulosa cells intercalated between hFGCs inside ovarian cords. The stromal compartment and ovarian cords are separated by white dotted lines. FOXL2 marks (pre)granulosa cells. **(D)** Immunofluorescence for YAP/panKRT/FOXL2 in an 18 WPF ovary. Dashed box is shown magnified at the bottom. The stromal compartment and ovarian cords are separated by white dotted lines. panKRT marks (pre)granulosa cells. **(E)** Uniform manifold approximation and projections (UMAP) plot showing cluster identity (CL_ID) and age distribution (WPF) of the gonadal somatic cells. **(F)** UMAP plots depicting the expression levels of genes of interest. **(G)** Whole-mount immunofluorescence for KRT19/NOTCH3/DDX4 in a 17 WPF ovary. White asterisks mark primordial follicles. Ovarian cords are separated by yellow dotted lines. The merged and single channels are shown. **(H)** Immunofluorescence for panKRT/JAG1 in 20 WPF (left panel) and UMAP plot showing *JAG1* expression in ovarian cells (right panel). Related to Fig S3D.

that YAP was localized to the cytoplasm of (mesenchymal) stromal cells as expected, but also to the cytoplasm of panKRT+FOXL2+ pre-granulosa cells (Fig 3D), indicative of active HIPPO signaling and epithelial–mesenchymal state of human pre-granulosa cells.

Next, we investigated the expression of specific keratins and observed higher levels of KRT19 in the outer cortex, whereas KRT8 and KRT18 were expressed in FOXL2+ (pre)granulosa cells, including PFs (Fig S3E). This analysis proved insufficient to distinguish between FOXL2+ pre-granulosa cells in ovarian cords and bona fide FOXL2+ granulosa cells in PFs. To resolve this, we extracted the cluster of *WT1+COL3A1+* gonadal somatic cells (186 cells) from our transcriptomics analysis (Fig S3C and D) and we

performed further sub-clustering to distinguish between sub-populations of granulosa cells. From the obtained five sub-clusters (CL), cells in CL0 and CL3 showed the expression of *TAGLN2* and low expression of *FOXL2*, suggesting that they corresponded to gonadal stromal cells (Fig 3E and F). Notably, part of CL0 showed high expression of *PODXL*, indicative of the ovarian surface epithelium. CL3 contained cells expressing high levels of *FKBP4*, indicative of stressed somatic gonadal cells (Fig 3E and F).

CL1, CL2, and CL4 contained cells expressing high levels of *FOXL2+* presumably of the granulosa cell lineage. From those clusters, CL4 expressed markers *NOTCH3*, *PCDH8*, *APOE*, and *COL4A4* (Fig 3F). Recently, Garcia-Alonso and colleagues also reported specific expression of *NOTCH3* in granulosa cell lineage at 17 WPF (Garcia-Alonso et al, 2022). Validation of NOTCH3 expression with whole-mount immunofluorescence confirmed its specificity in granulosa cells in PFs at 17 WPF, but not in pre-granulosa cells in ovarian cords surrounding meiotic hFGCs (Fig 3G). Furthermore, we confirmed the specific

expression of NOTCH ligand JAG1 in oocytes in PFs (Fig 3H) and high expression of *JAG1* in ZP3+ oocytes in PFs (Figs 3H and S3D). In humans, we observed NOTCH2 in meiotic hFGCs (Fig S3F) and NOTCH3/*NOTCH3* in granulosa cells in PFs, suggesting that the NOTCH receptor may be involved in the regulation of follicle formation in humans, as in mice (Trombly et al, 2009).

## Cadherin expression dynamics during the transition from ovarian cords to PFs

The presence of cell-membrane protrusions from pre-granulosa cells intercalated between meiotic hFGCs, the expression of CDH1 in FOXL2+ pre-granulosa cells at 12–16 WPF (Fig 3C), and the previously reported shift between *CDH2* and *CDH1* in meiotic hFGCs (Fan et al, 2021), led us to examine the cadherin-expression dynamics during the process of human PF formation between 18–20 WPF. We

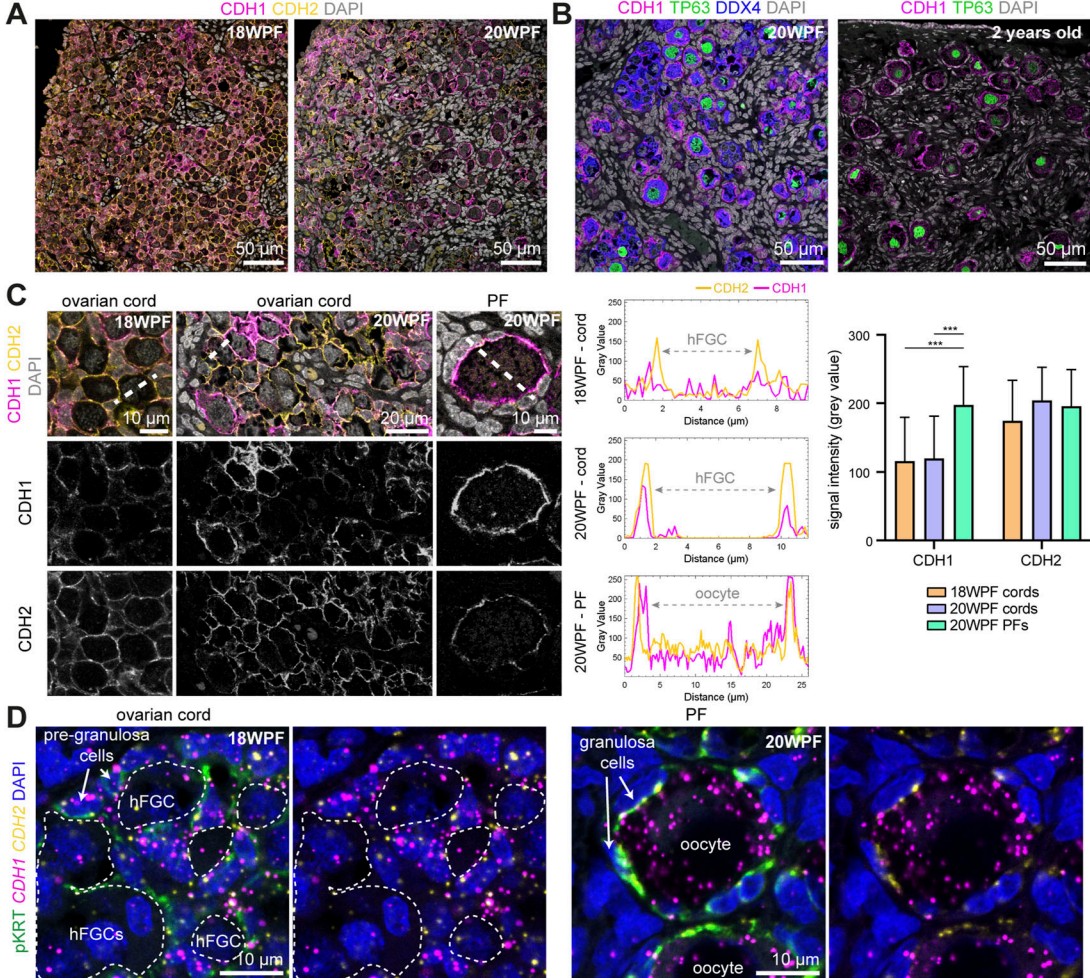

**Figure 4. Dynamics of cadherin expression during the formation of primordial follicles.**
**(A)** Immunofluorescence for CDH1/CDH2 in 18 and 20 WPF ovaries. **(B)** Immunofluorescence for CDH1/TP63/DDX4 in a 20 WPF ovary (left panel) and immunofluorescence for CDH1/TP63 in a 2-yr-old ovary (right panel). **(C)** Quantification of CDH1 and CDH2 expression at the interface between human fetal germ-cells/oocytes and (pre) granulosa cells in 18 and 20 WPF ovaries. Representative images for CDH1/CDH2 in 18 and 20 WPF ovaries, with single fluorescence channels used for quantification (region used depicted as dashed line) (left panels); with associated plots of grey values for CDH1 and CDH2 per cell (middle panels); and plot depicting signal intensity (mean ± SD) (right panel). (***$P$ < 0.001). **(D)** Confocal images of RNA FISH for *CDH1* and *CDH2* combined with immunofluorescence for pKRT in ovarian cords at 18 WPF and primordial follicles (PF) at 20 WPF.

detected both CDH1 and CDH2 at the interface between (pre) granulosa cells and hFGFs in ovarian cords and PFs (Fig 4A). Moreover, high CDH1 expression was maintained in PFs in a 2-yr-old ovary (Fig 4B). Quantification analysis of the levels of immunofluorescence of CDH1 and CDH2 at the interface between (pre) granulosa cells and hFGCs revealed that the deposition of CDH1 increased during the transition from ovarian cord to PF, whereas CDH2 levels remained comparable (Fig 4C). We performed RNA FISH for *CDH1* and *CDH2* and detected a pronounced increase in *CDH1* in oocytes during the transition to PFs, whereas *CDH2* was lowly expressed by both hFGCs and pKRT+ (pre)granulosa cells (Fig 4D). The up-regulation of CDH1 may lead to the formation of strong adhesive bonds between the

oocyte and associated (pre)granulosa cells facilitating the segregation of the formed PF.

## Establishment of cell–cell junctions during the transition from ovarian cords to PFs

Regarding the formation of tight junctions (TJ) during the transition between ovarian cords and PFs, the expression of TJ protein TJP1 significantly increased between 16–18 WPF at the contact points between the cell membrane of hFGCs and (pre)granulosa cells and remained high in PFs at 20 WPF (Fig 5A). Moreover, the presence of TJ between granulosa cells and the oocyte in PFs was validated using TEM (Fig 5B).

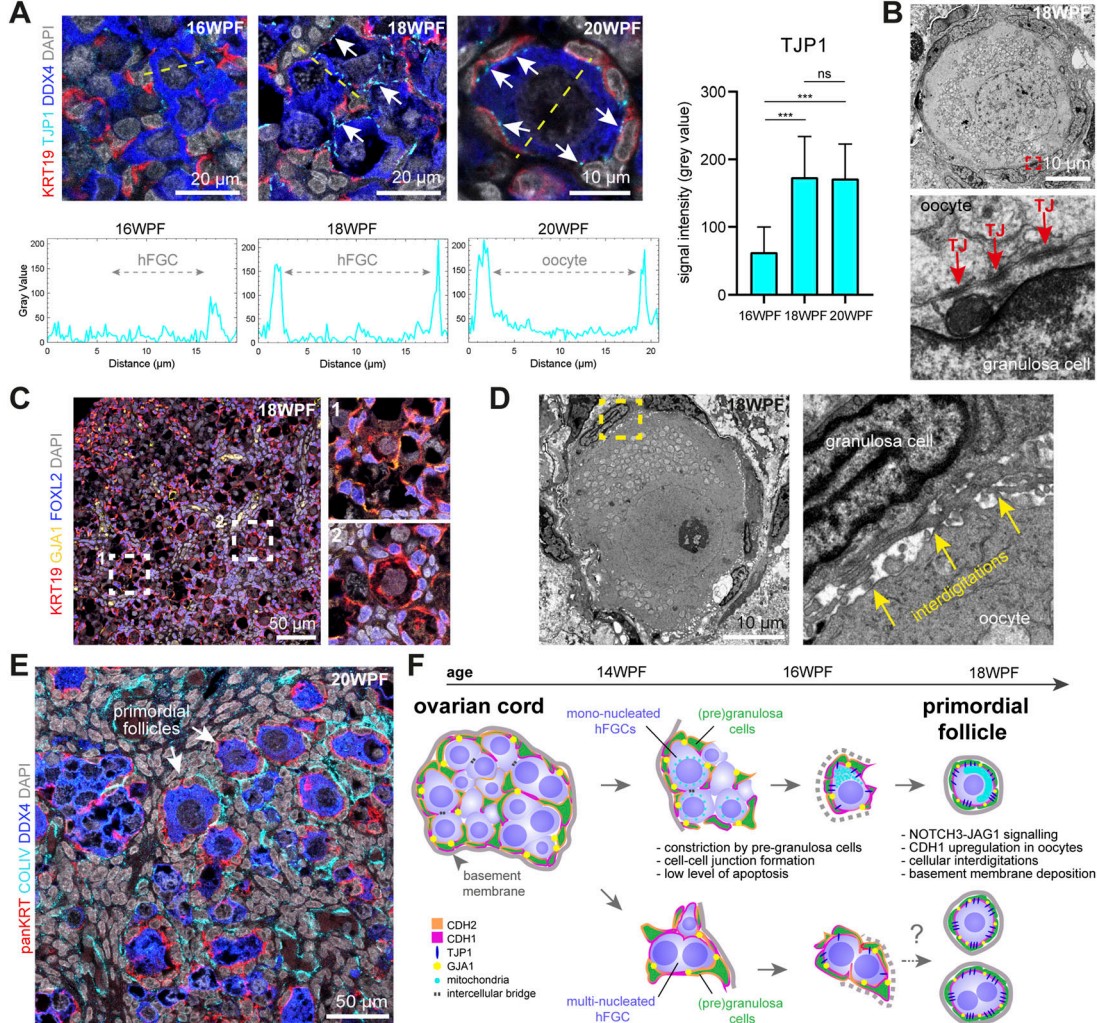

**Figure 5. Cell–cell interactions during the formation of primordial follicles.**
**(A)** Quantification of TJP1 expression at the interface between human fetal germ cells (hFGCs)/oocytes and (pre)granulosa cells in 16, 18, and 20 WPF ovaries. Representative images for TJP1/KRT19/DDX4 immunofluorescence used for quantification (region used is depicted by dashed lines) and white arrows point to TJP1+ tight junctions (TJ) (left top panels); with associated plots of grey values for TJP1+ per cell (left bottom panels); and plot depicting signal intensity (mean ± SD) (right panel). (***$P < 0.001$; ns, not significant). **(B)** Transmission electron microscopy images showing TJ between the granulosa cells and the oocyte in a primordial follicle at 18 WPF. Dashed box is shown magnified at the bottom. Arrows point to TJ. **(C)** Immunofluorescence for KRT19/GJA1/FOXL2 in an 18 WPF ovary. Dashed boxes are shown magnified to the bottom. FOXL2 marks (pre)granulosa cells. **(D)** Transmission electron microscopy images showing cellular interdigitations between the granulosa cells and the oocyte in a primordial follicle at 18 WPF. Dashed box is shown magnified at the right. Arrows point to the interdigitations. **(E)** Immunofluorescence for panKRT/COLIV/ DDX4 in a 20 WPF ovary. DDX4 marks hFGCs and oocytes. **(F)** Cartoon depicting a model for the transition between ovarian cord and primordial follicle in humans from mononucleated hFGCs and hypothetical for multinucleated hFGCs.

On the formation of gap junctions, we investigated the expression of GJA1, strongly expressed by granulosa cells in adult ovaries (Fan et al, 2019). We observed the expression of GJA1 in FOXL2+ pre-granulosa cells in ovarian cords and granulosa cells in PFs; however, a pronounced redistribution between these two stages was not observed (Fig 5C).

In addition to focal TJ between the oocyte and granulosa cells in PFs, we also observed the presence of cellular interdigitations between these two cell types using TEM (Fig 5D), as previously reported in human adult unilaminar follicles (Camboni et al, 2008; Woods et al, 2018), revealing an expansion of the surface area available to promote cell–cell adhesion and physically interlocking the oocyte and the granulosa cells together. Finally, to reinforce the PF structure, the (pre)granulosa cells deposit a basement membrane containing COLIV (Fig 5E) and LAM (Heeren et al, 2015), separating the PF from the surrounding stromal cells.

## Discussion

In this study, we investigated the cell–cell interactions between hFGCs and (pre)granulosa cells that take place during the second trimester during the transition from ovarian cords to PFs (Fig 5F). We reported that hFGCs develop either as single mononucleated hFGCs or as multinucleated hFGC–syncytia in the absence or presence of intercellular bridges, wide enough to allow mitochondria to pass through, illustrating not only heterogeneity in developmental stages, but also diversity in cytokinesis. Our results are in agreement with reports from mouse FGCs, where FGCs connect with neighboring FGCs, via intercellular bridges (on average 1.1–1.2 $\mu$m) to enable the transfer of organelles and cytoplasm necessary for proper germ-cell development (Lei & Spradling, 2016; Ikami et al, 2021). As in mice, apoptosis may not be the prevalent form of programmed cell death during the formation of PFs (Niu & Spradling, 2022). Moreover, some aspects were species-specific, for example, in contrast to mice (Pepling & Spradling, 2001; Niu & Spradling, 2022), in humans, the developmental progression of multinucleated hFGC–syncytia was not synchronized and the hFGC–syncytia were smaller regarding the number of connected hFGCs.

Given the presence of intercellular bridges and multinucleated hFGC–syncytia, we hypothesize that failure in the formation or maintenance of those bridges could lead to the generation of hFGC-syncytia without intercellular bridges. During cell division in somatic cells, duplicated chromosomes are attached to centrosomes (TUBG1+) via microtubules. If cytokinesis is successful, each daughter cell inherits one set of chromosomes and one centrosome. If this process fails in a cell that has completed karyokinesis, it will result in a binucleated cell with two centrosomes (Lens & Medema, 2019). Looking at the number of centrosomes in multinucleated and mononucleated hFGCs might shed light into the mechanism of the emergence of the multinucleation. In addition, performing live-imaging analysis of hFGC with labeled membranes and chromosomes would allow in-depth analysis of this phenomenon. However, these experiments are challenging because of the use of primary material, absence of suitable culture protocols, and phototoxicity. Finally, we stress that the formation of PFs from multinucleated hFGCs is hypothetical. It remains to be uncovered whether multinucleated hFGC–syncytia in humans function in a similar way to nurse FGCs–oocytes described in mice (Niu & Spradling, 2022), even though they are capable of entering meiosis, rather than becoming future PFs.

Previous studies that investigated the mechanism of PF formation focused either on signaling pathway analysis or the role of specific genes in this process (Paredes et al, 2005; Bristol-Gould et al, 2006; Hutt et al, 2006; Nicol et al, 2020); however, when and how hFGCs and pre-granulosa cells establish physical interactions to form PFs have not been explored. We report that in humans, pre-granulosa cells already at 12 WPF formed thin cytoplasmic protrusions that engulf individual hFGCs, impeding direct contact with neighboring hFGCs, perhaps in preparation for follicle formation. What was more surprising, pre-granulosa cell protrusions were even constricting into meiotic multinucleated hFGCs–syncytia. Encapsulation and constriction of hFGCs by pre-granulosa cells in the transition from ovarian cord into PF in humans may be a comparable process with that described in *Drosophila* during germ cell cyst formation (Chanet & Huynh, 2020). It remains unclear whether this constriction mechanism leads to follicular formation, preventing the formation of follicles with multinucleated oocytes.

The concerted expression of VIM and CDH1 together with pERM inside the ovarian cords may drive changes in cell rigidity and cell–cell adhesion, whereas facilitating the formation, positioning, and maintenance of cell protrusions by the pre-granulosa cells (Morales et al, 2004; Helfand et al, 2011; Collins et al, 2017) guiding the attachment to specific hFGCs in preparation to follicular formation. When cells express high levels of the same type of cadherins (homophilic interaction), they adhere strongly and undergo cell sorting, whereas cells expressing different types of cadherins form weaker cell–cell adhesion (Maitre et al, 2012). Therefore, the up-regulation of CHD1 in oocytes together with CDH1 expression in pre-granulosa cells will allow the formation of strong homophilic bonds between the two cell types. In addition, cellular interdigitations have been shown to contain high levels of CDH1, complementing physical interlocking with homophilic cell–cell adhesion (Li et al, 2021). Together with the establishment of TJs (TJP1+) and gap junctions (GJA1+), all these mechanisms reinforce physical anchors between hFGCs and (pre)granulosa cells, whereas COLIV deposition around the PF seals the follicular structure.

In conclusion, our findings highlight the complex developmental strategy used in humans to establish physical interactions between the germline and the soma long before PFs are formed. We propose a model of PF formation in humans (Fig 5F) whereby the NOTCH pathway, an interplay of cell–cell signaling, cell–cell adhesion, and physical interactions between hFGCs and (pre)granulosa cells play important roles in formation of PFs, which form the finite pool of oocytes during an entire woman's life.

## Materials and Methods

### Ethical statement and sample collection

The collection and use of human fetal material (12–20 WPF) received a letter of no objection from by the Medical Ethical

Committee of the Leiden University Medical Center (P08.087 and B21.052). Human material was obtained from elective abortions (without medical indication) and donated for research purposes with written informed consent. The developmental age was determined before the procedure by obstetric ultrasonography. Paraffin sections of two late fetal (30 and 38 WPF) and one child (2 yr old) ovaries were provided by the biobank of the Department of Pathology of the Leiden University Medical Center. Paraffin sections of one human adult testicular biopsy were obtained from an adult male undergoing testicular biopsy in search of spermatozoa to perform an intracytoplasmic sperm injection. The participant signed the informed consent, approved by the Basque Ethics Committee for Clinical Research (CEIC-E PI2014205).

### TEM and associated imaging

A small portion of the ovary from three different donors (17–18 WPF) was fixed for 1 h in 1.5% glutaraldehyde (EM grade; Electron Microscopy Sciences) in 0.1 M cacodylate buffer (pH 7.4) (A2140; PanReac AppliChem) at RT. After rinsing the tissue three times in 0.1 M cacodylatebuffer (Sigma-Aldrich), it was postfixed in a solution of 1% $OsO_4$ (Electron Microscopy Sciences), 1.5% potassiumhexacyanoferrate (III) (Merck), and 0.1 M cacodylate solution for 1 h on ice. Next, the tissue was dehydrated in 70% EtOH (VWR chemicals, absolute ethanol) overnight (o/n) at 4°C. The next day, the tissue was further dehydrated in 80% and 90% EtOH for 10 min each, and twice in 100% EtOH for 30 min. The tissue was then impregnated in mixtures of propylene oxide (Sigma-Aldrich) and EPON (LX112; Ladd research industries) (2:1, 1:1, and 1:2), for 30 min each followed by pure EPON for 60 min at RT. The tissue was positioned in an embedding mold (BEEM) and EPON was added. EPON was allowed to harden for 2 d at 70°C in an oven (Dépex).

Initially semi-thin slices were cut, pasted on a glass slide (Starfrost, Knittel Glaser), and stained with a solution of 1% toluidine blue (Merck) allowing light microscopic evaluation. After this, areas of interest were selected and sliced using an Element Six ultramicrotome knife (Drukker) in a Reichert Ultracut S microtome (Leica). Ultrathin slices (80 nm) were mounted on copper slotgrids (Vecro, Slotgrids) before being stained with a 20-$\mu$l solution of 7% uranyl acetate (Fluka) in the dark at RT for 10 min. The sections were then washed 10 times in milliQ water, twice with 0.01 M in NaOH (JT Baker), and stained for 5 min with 20 $\mu$l a solution of Reynolds lead citrate (Santhana Raj et al, 2021) and washed 10 times in 0.01% NaOH and 10 times in milliQ water before being air-dried and imaged with a Twin electron microscope (Tecnai T12Twin; Fei). Samples were imaged with a Gatan camera (One View) that stitched several photographs together. Stitches were analyzed using Aperio ImageScope (version 12.3.2.8013; Leica) (Faas et al, 2012).

Existing TEM images from mice fetal ovaries (Ikami et al, 2021) were reanalysed for this study.

### Histology and immunofluorescence

Ovarian tissue was fixed, embedded in paraffin, sectioned, and processed for immunofluorescence as described previously (Heeren et al, 2015). For antigen retrieval, tissue sections were treated for 20 min at 98°C in a microwave (TissueWave 2; Thermo

Fisher Scientific) with TRIS–EDTA pH 9.0 (10 mM Tris [Sigma-Aldrich] and 1 mM EDTA [Life Technologies]) and allowed to cool to RT. Sections were treated with a blocking solution of 1% BSA (Sigma-Aldrich) in PBST 0.05% Tween-20 (Merck) in PBS at RT for 1 h and incubated with primary antibodies (Table S1) diluted in the blocking solution on at 4°C in a humidified chamber. Next day, the slides were washed twice in 1x PBS for 5 min, once in PBST for 5 min, and incubated for 1 h at RT with secondary antibodies (Table S1) and (DAPI, dilution 1:1,000; Thermo Fisher Scientific) diluted in the blocking solution in a humidified chamber. All slides were mounted with ProLong Gold (Thermo Fisher Scientific).

### Whole-mount immunofluorescence

After isolation, second trimester fetal ovaries were fixed o/n at 4°C in 4% PFA (Merck). The following day, the tissue was washed three times for 30 min in 1x PBS and stored in 70% EtOH at 4°C. Thereafter, the tissue was cut into 80-$\mu$m-thick slices using vt1200 S vibratome (Leica) or in small pieces using tungsten needles. Tissue slices were permeabilized for 20 min in 0.3% TritonX-100 (Sigma-Aldrich) in PBS and treated o/n at 4°C with a blocking solution containing 1% BSA and 0.3% TritionX-100 in PBS on a shaker. Thereafter, the tissue slices were incubated with primary antibodies (Table S1) in the blocking solution at 4°C on a shaker for 2 d. Next, the tissue slices were washed at RT in 0.2% TritonX-100/PBS for 2 h and incubated with secondary antibodies (Table S1) and DAPI o/n at 4°C. Slices were washed in 0.2% TritionX-100/PBS for 2 h and placed on a $\mu$-Slide 18-well ibiTreat slides (IBIDI) or in 35-mm glass-bottom dishes (MatTek Corporation) for confocal imaging.

### Combined RNA FISH and immunofluorescence

RNA FISH in combination with immunofluorescence was performed on paraffin sections as previously described (Fan et al, 2021). Briefly, the RNAscope Multiplex fluorescent reagent kit v2 (323100; Advanced Cell Diagnostics) was used with either RNAscope Probe-Hs-TEX14-C1 (1255731-C1) or RNAscope Probe-Hs-CDH1-C3 (311091-C3) with Probe-Hs-CDH2-C2 (310171-C2) on a HybEZ II oven (321720; Advanced Cell Diagnostics).

After hybridization, the TEX14 probe was amplified with Opal 520 dye (1:1,000, FP1487001KT; Akoya Biosciences). Sections were then blocked with 10% normal horse serum (S-2000; Vector Laboratories), incubated with primary antibody rabbit anti-pERM (1:200) (Table S1) o/n at 4°C, followed by incubation for 30 min at RT with HRP-linked swine anti-rabbit immunoglobulins (1:200) (Table S1), treated with Opal 690 dye (1:1,500, FP1497001KT; Akoya Biosciences), counterstained with DAPI, and mounted with ProLong Gold (Life Technologies).

After hybridization with the CDH1 and CDH2 probes (1:50 in probe dilutent), the mRNA signal was amplified sequentially for each channel. Fluorophores used to detect signals were Opal 570 dye (1:2,000, FP1488001KT; Akoya Biosciences) for CDH1 and Opal 690 dye (1:1,500) for CHD2. Next, the sections were blocked with 10% normal horse serum, incubated with primary antibody mouse anti-panKRT (1:100) (Table S1) o/n at 4°C, followed by HRP-linked goat anti-mouse immunoglobulins (1:1,000) (Table S1), treated with diluted

Opal 520 dye (1:1,000), counterstained with DAPI, and mounted with ProLong Gold.

### Confocal imaging, image quantification, and statistical analysis

Confocal images were obtained on an inverted SP8 confocal microscope (Leica) with LAS X software (Leica) with the 40x oil immersion objective (HC PL APO 40x/1.40 Oil) and 63x oil immersion objective (HC PL APO 63x/1.30 Oil), and Andor Dragonfly 500 spinning disk microscope (Oxford Instruments) with the 63x oil immersion objective (HC PL APO 63x/1.30 Oil). Confocal images from an SP8 microscope were processed in ImageJ (Schindelin et al, 2012); spinning-disk images were processed in Imaris 9.5.0 (Bitplane) software. Figures were assembled using Illustrator CC 2020 (Adobe).

For the quantification of mono and multinucleated hFGCs at 14–17 WPF (N = 4) and 18–20 WPF (N = 4), at least 150 DDX4+ hFGCs per image were counted on a single z-plane in DDX4/CTNNB1 and DDX4/pERM immunofluorescence images. Only hFGCs with visible nuclei and intact cell membranes were counted.

For the quantification of the signal intensity for CDH1 and CDH2, the interface between hFGCs/oocytes and two neighboring (pre) granulosa cells was measured (in arbitrary units or gray values) in at least 100 hFGCs/oocytes at 18 and 20 WPF using the "Plot Profile" function in ImageJ software on confocal images. The two highest values, corresponding to the two regions of interface, corresponding to the intensity of CDH1 and CDH2 were measured in the same cells. A similar approach was used to quantify the signal intensity of TJP1 at the interface between hFGCs/oocytes and two neighboring (pre)granulosa cells at 16, 18, and 20 WPF.

The quantification results, presented as mean ± SD, were analyzed with GraphPad Prism v9.0.1 software (Graph Pad Software Inc.). Statistical significance was determined using two-way ANOVA (CDH1, CDH2) and one-way ANOVA (TJP1). $P$-value < 0.05 (*), < 0.01 (**), and < 0.001 (***) were considered statistically significant.

### Quantification of intercellular bridge diameter

The diameter of intercellular bridges between hFGCs was measured in TEM images from three independent samples at the widest point using "Ruler" tool in Aperio ImageScope (version 12.3.2.8013; Leica) (Faas et al, 2012). The graph was made using R (v4.1.0).

### Single-cell RNA-sequencing data analysis

The transcripts per million-normalized count tables of human fetal gonad single-cell RNA-sequencing dataset were downloaded from Gene Expression Omnibus (GEO) database with accession number GSE86146 (Li et al, 2017). The data extraction, processing, and quality control of the female gonadal cells from 18–26 WPF were done as previously described (Fan et al, 2021). The dataset was analyzed using a Seurat-based workflow (v4.0.3) (Hao et al, 2021) using R (v4.1.0). For cell clustering, the 12 first principal components were used with resolution parameter setting 0.8 and k.param setting 7. The same parameters were kept for sub-clustering the ovarian somatic cells. Violin plot was generated using function "VlnPlot" in the Seurat workflow.

## Data Availability

The dataset analyzed in this study is available in the following database: RNA-seq data: Gene Expression Omnibus GSE86146.

## Supplementary Information

## Acknowledgements

We would like to thank Dr. T Bosse (Dept. of Pathology, LUMC), Dr. C Jost (Dept. of Cell Chemical Biology, LUMC), and the members of the SM Chuva De Sousa Lopes group for insightful discussions; J Taelman and E Lam for technical support; and the teams of the abortion clinic Vrelinghuis in Utrecht and Gynaikon Klinieken in Rotterdam for assistance in providing tissue samples and the anonymous volunteer donors that made this study possible. This study was funded by the European Research Council (ERC-CoG-2016-725722 OVOGROWTH) to SM Czukiewska, X Fan, and SM Chuva De Sousa Lopes; the Dutch Research Council (VICI-2018-91819642) to SM Chuva De Sousa Lopes; the Dutch Organization for Health Research and Development (ZonMw PSIDER 10250022120001) to T Van Der Helm and SM Chuva De Sousa Lopes; the China Scholarship Council (CSC 201706320328) to X Fan; the National Institute of General Medical Science (R01GM126028) to L Lei; and the Novo Nordisk Foundation grant (reNEW NNF21CC0073729) to SM Czukiewska, S Hillenius, and SM Chuva De Sousa Lopes.

### Author Contributions

SM Czukiewska: conceptualization, data curation, formal analysis, validation, investigation, visualization, methodology, and writing—original draft, review, and editing.
X Fan: conceptualization, data curation, formal analysis, validation, investigation, visualization, methodology, and writing—original draft, review, and editing.
AA Mulder: resources, data curation, formal analysis, investigation, visualization, methodology, and writing—original draft, review, and editing.
T Van Der Helm: resources, formal analysis, validation, investigation, visualization, methodology, and writing—original draft, review, and editing.
S Hillenius: resources, formal analysis, validation, investigation, visualization, methodology, and writing—original draft, review, and editing.
L Van Der Meeren: resources, formal analysis, investigation, methodology, project administration, and writing—original draft, review, and editing.
R Matorras: resources, formal analysis, investigation, methodology, project administration, and writing—original draft, review, and editing.
C Eguizabal: resources, formal analysis, investigation, methodology, project administration, and writing—original draft, review, and editing.
L Lei: data curation, formal analysis, validation, investigation, visualization, methodology, and writing—original draft, review, and editing.

RI Koning: resources, data curation, formal analysis, investigation, visualization, methodology, and writing—original draft, review, and editing.

SM Chuva De Sousa Lopes: conceptualization, resources, data curation, formal analysis, supervision, funding acquisition, validation, investigation, visualization, methodology, project administration, and writing—original draft, review, and editing.

## Conflict of Interest Statement

The authors declare that they have no conflict of interest.

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
