## [Reviewer comments · Life Science Alliance]

Life Science Alliance

Cell-cell interactions during formation of primordial follicles in humans

Sylwia CZUKIEWSKA, Xueying FAN, Aat Mulder, Talia VAN DER HELM, Sanne HILLENUS, Lotte VAN DER MEEREN, Roberto MATORRAS, Cristina Eguizabal, Lei LEI, Roman Koning, and Susana Chuva de Sousa Lopes

DOI: <https://doi.org/10.26508/lsa.202301926>

Corresponding author(s): *Susana Chuva de Sousa Lopes, Leiden University Medical Center*

Review Timeline:	Submission Date:	2023-01-14
	Editorial Decision:	2023-03-06
	Revision Received:	2023-07-26
	Editorial Decision:	2023-08-17
	Revision Received:	2023-08-19
	Accepted:	2023-08-21

Scientific Editor: Novella Guidi

Transaction Report:

March 6, 2023

Re: Life Science Alliance manuscript #LSA-2023-01926-T

Prof. Susana M Chuva de Sousa Lopes
Leiden University Medical Center
Dept. of Anatomy and Embryology
Einthovenweg 20, 2300 ZC Leiden
Leiden 2300 RC Leiden
Netherlands

Dear Dr. Chuva de Sousa Lopes,

Thank you for submitting your manuscript entitled "Cell-cell interactions regulate the formation of primordial follicles in humans" to Life Science Alliance. The manuscript was assessed by expert reviewers, whose comments are appended to this letter. We invite you to submit a revised manuscript addressing the Reviewer comments.

Thank you for this interesting contribution to Life Science Alliance. We are looking forward to receiving your revised manuscript.

Sincerely,

B. MANUSCRIPT ORGANIZATION AND FORMATTING:

Reviewer #1 (Comments to the Authors (Required)):

The manuscript by Czukiewska and colleagues describes the formation of interconnected germline cysts in human females. This process takes place before birth and the availability of such tissues is thus scarce. Given the human origin of the material, the study is entirely descriptive, but it is a very nice description. The images are stunning and despite the complexity of the tissues, it is possible for non-specialists to distinguish the different cell types. The nomenclature could be made easier (see below) for non-specialist readers.

My main criticisms regard the phrasing of the main conclusions. The claims are not supported by experimental data, so I would advise to be much softer on the conclusions. For example:

- 1) Title: "Cell-cell interactions regulate the formation of primordial follicle in humans". There may be cell-cell interactions because adhesion molecules are expressed but the authors do not show whether these proteins are actually required for cell-cell adhesions, and even less that these interactions are required for proper formation of a primordial follicle. The title of the manuscript cannot be that assertive.
- 2) Abstract: "This process involved a cadherin-regulated cell sorting". It is not demonstrated anywhere. There is just a simple correlation of cadherin being expressed in these cells and the possibility of cell sorting happening at this time.
- 3) Graphical abstract: the upper pathway shows multinucleated cells giving rise to single cell primordial follicle. Again this is not demonstrated anywhere in the manuscript.

These are just a few examples of conclusions not-supported by data in the most critical parts of the manuscript, but there are other instances in the text.

Other comments:

- 1) It is hard to follow what all the antibodies are labelling in the different figures. Instead of using systematically protein names, would it be possible to use simple descriptive labels such as soma, germline, meiotic, mitotic, apoptosis, etc...?
- 2) KIF23 labels canals but it could also mark midbody remnants. Could it be possible to use specific intercellular bridge markers such as TEX14?
- 3) Regarding TEX14: is it expressed in human cysts? If not, could the authors compare wild type human cysts with tex14 mutant mouse cysts? And more specifically, the formation of germline cysts without stable intercellular bridges.
- 4) In figure 2C, it is not obvious where the cytoplasmic bridge is (despite the blue dotted line). The authors could mark the two connected cells as cell#1 and cell#2.
- 5) Centrosomes can move between cells through the canals, so counting the number of centrosomes may not be the most accurate way to detect cytokinesis failure.
- 6) The nuclear localization of TJP1 in the nucleus is surprising but not very informative in the context of this manuscript. Would it be possible to use another junction marker?
- 7) It is not clear whether CDH1 and CDH2 are expressed in germline or somatic cells or both. The authors could perform RNA FISH for CDH1 and CDH2 to detect in which cell types they are expressed.

Reviewer #2 (Comments to the Authors (Required)):

Czukiewska SM et al. discovered the multinucleated syncytia formed by hFGC in human cortical tissues, which is associated with intercellular bridges. They observed the interaction between hFGCs and pre-granulosa cells during the formation of primordial follicles in human fetal tissues. This paper provided high-quality staining images and TEM images to characterize the properties of hFGCs and proposed a model for studying PF formation in human tissues. However, this manuscript could be improved in these aspects.

1. Please check the 'manuscript format' of LSA. The main manuscript should include an appropriate Introduction with current knowledge of this field and the rationale of this study, Results describing the findings from this study with proper subtitles, and a Discussion. The authors might want to reorganize and rephrase the main text of this manuscript according to the guidance of LSA.
2. There are plenty of abbreviations that need to add full names in this manuscript. E.g. NANOG, PDPN, ALPL, POU5F1, etc.

The authors should include the full name of these abbreviations properly.

3. The authors might want to expand the Introduction session with more information on gametogenesis, hFGC development, regulation of cellular interactions, EMT and its related pathways, etc.
4. Figure 1, is there a reason why the authors used different aged tissues (14, 16, 17, 18 WPF) for panels A-C? Can the authors provide the IF/TEM staining images from the same age to better compare the phenomenon?
5. Figure 1B & 1C, What's the percentage of multi-nucleated syncytia with/without intercellular bridges?
6. Figure 2B, are all hFGC-syncytia without intercellular bridges cCASP3 negative? Please provide the quantification data with the % of cCASP3+/- populations.
7. Figure 2D did the authors observe a decrease/increase in the diameter of bridges by age? Can the authors compare the diameter of bridges from 12-20 WPF as you obtained?
8. Figure 2F, can the authors provide the % of syncytia containing two centrosomes?
9. Figure 3C could be moved to supplemental material, as the data was insufficient to distinguish FOXL2+ pre-granulosa cells from FOXL2+ granulosa cells.
10. Line 165, please check and rephrase the sentence. The authors might want to delete the 'from'
11. Line 185, the authors might want to change 'is' to in
12. Figure 4A & 4B, please provide the intensity quantification analysis of CDH1 and CDH2 expression to support the statement in Lines 193-198.
13. Figure 4D, please quantify the expression of TJP1 between hFGCs and pre-granulosa cells at different ages.
14. Figure 4, did the authors notice any changes in organelle enrichment (mitochondrial, golgi, etc.) during the formation of tight junctions and cell-to-cell adhesion? To test this, the authors may want to analyze the number of Golgi/mitochondrial per cell in different aged human cortical.

Reviewer #1 (Comments to the Authors (Required)):

The manuscript by Czukiewska and colleagues describes the formation of interconnected germline cysts in human females. This process takes place before birth and the availability of such tissues is thus scarce. Given the human origin of the material, the study is entirely descriptive, but it is a very nice description. The images are stunning and despite the complexity of the tissues, it is possible for non-specialists to distinguish the different cell types. The nomenclature could be made easier (see below) for non-specialist readers. My main criticisms regard the phrasing of the main conclusions. The claims are not supported by experimental data, so I would advise to be much softer on the conclusions.

We thank the Reviewer #1 for a detailed and constructive feedback. We have followed most requested suggestions and rephrased the main conclusions, making them softer where appropriate throughout the text. Please see our point-by-point answers below:

Question 1. Title: "Cell-cell interactions regulate the formation of primordial follicle in humans". There may be cell-cell interactions because adhesion molecules are expressed but the authors do not show whether these proteins are actually required for cell-cell adhesions, and even less that these interactions are required for proper formation of a primordial follicle. The title of the manuscript cannot be that assertive.

Answer 1. We agree with the Reviewer and we modified the title to "Cell-cell interactions during formation of primordial follicles in humans"

Question 2. Abstract: "This process involved a cadherin-regulated cell sorting". It is not demonstrated anywhere. There is just a simple correlation of cadherin being expressed in these cells and the possibility of cell sorting happening at this time.

Answer 2. We agree that referred statement is not supported by sufficient evidence therefore we have removed that statement from the Abstract.

Question 3. Graphical abstract: the upper pathway shows multinucleated cells giving rise to single cell primordial follicle. Again this is not demonstrated anywhere in the manuscript.

Answer 3. As suggested by the Reviewer, we have now modified the Graphical Abstract, emphasizing the hypothetical part of the multinucleated cells (now at the bottom, including a question mark and a multinucleated primordial follicle as well).

Page 18 line 539-540: In the legend of Fig 5F we now also state: "(F) Cartoon depicting a model for the transition between ovarian cord and primordial follicle in humans from mono-nucleated hFGCs and hypothetical for multi-nucleated hFGCs."

These are just a few examples of conclusions not-supported by data in the most critical parts of the manuscript, but there are other instances in the text.

We thank the Reviewer for this valid comment and we have moved several parts of the manuscript to the novel Discussion part and or made them less strong throughout the main text.

Other comments:

Question 4. It is hard to follow what all the antibodies are labelling in the different figures. Instead of using systematically protein names, would it possible to use simple descriptive labels such as soma, germline, meiotic, mitotic, apoptosis, etc...?

Answer 4. As suggested, we now indicate in the Figure legends which cell types are marked by immunofluorescence. We hope that this improves clarity and help readers understand which cell types are discussed.

Question 5. KIF23 labels canals but it could also mark midbody remnants. Could it be possible to use specific intercellular bridge markers such as TEX14?

Answer 5. We thank the Reviewer for this suggestion. Currently, there is no commercially available TEX14 antibody that recognizes human TEX14 protein. Therefore, to follow Reviewer's suggestion we evaluated *TEX14* expression using a single cell transcriptomics dataset (**new Fig 2E**). In addition, to further verify *TEX14* expression we opted for RNA FISH for *TEX14* and confirmed that *TEX14* RNA can be found in pre-meiotic and meiotic hFGCs (**new Fig 2F**).

Page 6 line 153-156: *"As expected, TEX14 was not expressed in the gonadal somatic and immune cells, but was highly expressed in meiotic hFGCs (Fig 2E). In agreement, expression of TEX14 was also confirmed by RNA fluorescent in situ hybridization (FISH) in pERM+ hFGCs (Fig 2F)."*

Question 6. Regarding TEX14: is it express in human cysts? If not, could the authors compare wild type human cysts with *tex14* mutant mouse cysts? And more specifically, the formation of germline cysts without stable intercellular bridges.

Answer 6. Regrettably it is impossible for us to compare the *TEXT14* expression in wild type human cysts with *Tex14* mutant mouse cysts.

Question 7. In figure 2C, it is not obvious where the cytoplasmic bridge is (despite the blue dotted line). The authors could mark the two connected cells as cell#1 and cell#2.

Answer 7. Following Reviewer's suggestion, we have labelled two connected cells as hFGC #1 and hFGC #2 in (new label) **Fig 2A** and we used blue arrows to better highlight where the intercellular bridges are. We hope this satisfies the Reviewer.

Question 8. Centrosomes can move between cells through the canals, so counting the number of centrosomes may not be the most accurate way to detect cytokinesis failure.

Answer 8. We were unable to performed the requested quantification due to the challenge of finding a sufficient number on multinucleated cells that recently finished dividing to quantify the number of centrosomes. Therefore, we have decided to remove the analysis of centrosomes from the Result section and only mention cytokinesis failure in the Discussion. See also Question 8 from Reviewer #2.

Page 9 line 256-264: *"Given the presence of intercellular bridges as well as multinucleated hFGC-syncytia, we hypothesize that failure in the formation or maintenance of those bridges could lead to the generation of hFGC-syncytia without intercellular bridges. During cell division in somatic cells, duplicated chromosomes are attached to centrosomes (TUBG1+) via microtubules. If cytokinesis is successful, each daughter cell inherits one set of chromosomes and one centrosome. If this process fails in a cell that has completed karyokinesis, it will result in a binucleated cell with two centrosomes (Lens & Medema, 2019). Looking at the number of centrosomes in multinucleated and mononucleated hFGCs might shed light into the mechanism of the emergence of the multinucleation."*

Question 9. The nuclear localization of TJP1 in the nucleus is surprising but not very informative in the context of this manuscript. Would it be possible to use another junction marker?

Answer 9. We are very thankful for this great suggestion! We have compared two different TJP1 antibodies and indeed found that the newly bought monoclonal TJP1 antibody (Life Technologies) only marked the tight junctions and not the meiotic nuclei of hFGCs. We have replaced all images with the correctly working TJP1 (**new Fig 5A**).

Question 10. It is not clear whether CDH1 and CDH2 are expressed in germline or somatic cells or

both. The authors could perform RNA FISH for CDH1 and CDH2 to detect in which cell types they are expressed.

Answer 10. Following the Reviewer's suggestion, we have now performed RNA FISH for CDH1 and CDH2 to identify which cells express them in ovarian cords and primordial follicles (**new Fig 4D**). In addition, as requested by Reviewer #2 (Question 12) we also quantified the protein expression of CDH1 and CDH2 at the junction between germline or somatic cells (**new Fig 4C**).

Page 8 line 219-221: *"We performed RNA FISH for CDH1 and CDH2 and detected a pronounced increase in CDH1 in oocytes during the transition to PFs, whereas CDH2 was lowly expressed by both hFGCs and pKRT+ (pre)granulosa cells (Fig 4D)."*

Reviewer #2 (Comments to the Authors (Required)):

Czukiewska SM et al. discovered the multinucleated syncytia formed by hFGC in human cortical tissues, which is associated with intercellular bridges. They observed the interaction between hFGCs and pre-granulosa cells during the formation of primordial follicles in human fetal tissues. This paper provided high-quality staining images and TEM images to characterize the properties of hFGCs and proposed a model for studying PF formation in human tissues. However, this manuscript could be improved in these aspects.

We thank Reviewer #2 for the recommendations and time invested in the manuscript to improve its quality.

Question 1. Please check the 'manuscript format' of LSA. The main manuscript should include an appropriate Introduction with current knowledge of this field and the rationale of this study, Results describing the findings from this study with proper subtitles, and a Discussion. The authors might want to reorganize and rephrase the main text of this manuscript according to the guidance of LSA.

Answer 1. We have reformatted manuscript structure throughout to conform to LSA guidance. Briefly, we have broadened the Introduction, including current knowledge of this field and the rationale of this study. Furthermore, we added proper subtitles to the Results and separated Results from Discussion.

Question 2. There are plenty of abbreviations that need to add full names in this manuscript. E.g. NANOG, PDPN, ALPL, POU5F1, etc. The authors should include the full name of these abbreviations properly.

Answer 2. We have defined all abbreviations in the main text.

Regarding gene nomenclature: through the main text we have used the standard nomenclature and official gene name or gene symbol to refer to genes/proteins. These are not abbreviations as such, but the current and unambiguously accepted nomenclature (see for example genecards.org or ensembl.org). Each gene can have many names, therefore there was a need to refer to genes by a standard unique gene symbol. We hope this explanation satisfies the Reviewer.

Question 3. The authors might want to expand the Introduction session with more information on gametogenesis, hFGC development, regulation of cellular interactions, EMT and its related pathways, etc.

Answer 3. As suggested by the Reviewer, we have expanded the Introduction section considerably with additional information on gametogenesis, hFGC development, regulation of cellular interactions, EMT and related pathways.

Question 4. Figure 1, is there a reason why the authors used different aged tissues (14, 16, 17, 18 WPF) for panels A-C? Can the authors provide the IF/TEM staining images from the same age to better compare the phenomenon?

Answer 4. We thank the Reviewer for this suggestion. For better comparison between the panels in Fig 1A-1C, we now only use samples from 17-18WPF to better compare this phenomenon.

Question 5. Figure 1B & 1C, What's the percentage of multi-nucleated syncytia with/without intercellular bridges?

Answer 5. We thank the Reviewer for this suggestion, but we think a quantification of bridges in TEM sections will result in a gross underestimation and is therefore not feasible. Nevertheless, we have now included a quantification of the multi-nucleated syncytia in paraffin sections (new Fig 1E), mentioning the technical limitation.

Page 8 line 219-221: *"We quantified the number of mono- and multinucleated DDX4+ hFGCs in 14-17WPF (N=4) and 18-20WPF (N=4) ovaries on paraffin sections and observed that about 7% of DDX4+ FGCs were present in multinucleated syncytia (Fig 1E), although this may be an underestimation, due to fact that we are counting on sections."*

Question 6. Figure 2B, are all hFGC-syncytia without intercellular bridges cCASP3 negative? Please provide the quantification data with the % of cCASP3+/- populations.

Answer 6. Following Reviewer's suggestion, we have now included the quantification data with percentages of cCASP3+/- populations (new Fig 2I).

Page 8 line 219-221: *"After quantification on paraffin sections, we showed that at 14-17WPF and 18-20WPF about 4% of the total number of DDX4+ FGCs were cCASP3+, with the majority corresponding to mononucleated hFGCs (Fig 2I). Although this may be underrepresented, this percentage of apoptosis is comparable to what has been reported in mice, where a different type of programmed cell death (by acidification) was reported (Niu & Spradling, 2022)."*

Question 7. Figure 2D did the authors observe a decrease/increase in the diameter of bridges by age? Can the authors compare the diameter of bridges from 12-20 WPF as you obtained?

Answer 7. In our manuscript, we measured bridge diameter in TEM images. Because we do not have TEM images of fetal ovaries younger than 17WPF, we cannot perform this analysis. It will definitely be important to look into this phenomenon and assess intercellular bridge dynamics/behavior.

Question 8. Figure 2F, can the authors provide the % of syncytia containing two centrosomes?

Answer 8. This would have been very interesting to investigate, but proved very difficult to quantify in paraffin sections, therefore, we have decided to remove the analysis of centrosomes from the Result section and only mention cytokinesis failure in the Discussion. See also Question 8 from Reviewer #1.

Page 9 line 256-264: *"Given the presence of intercellular bridges as well as multinucleated hFGC-syncytia, we hypothesize that failure in the formation or maintenance of those bridges could lead to the generation of hFGC-syncytia without intercellular bridges. During cell division in somatic cells, duplicated chromosomes are attached to centrosomes (TUBG1+) via microtubules. If cytokinesis is successful, each daughter cell inherits one set of chromosomes and one centrosome. If this process fails in a cell that has completed karyokinesis, it will result in a binucleated cell with two centrosomes (Lens & Medema, 2019). Looking at the number of centrosomes in multinucleated and mononucleated hFGCs might shed light into the mechanism of the emergence of the multinucleation."*

Question 9. Figure 3C could be moved to supplemental material, as the data was insufficient to distinguish FOXL2+ pre-granulosa cells from FOXL2+ granulosa cells.

Answer 9. As requested, images in Fig 3C were moved to Fig S3E.

Question 10. Line 165, please check and rephrase the sentence. The authors might want to delete the 'from'.

Answer 10. We have rephrased the sentence as suggested.

Page 6 line 148: *“To this end an online available single-cell transcriptomics dataset was used (Li et al., 2017)”*

Question 11. Line 185, the authors might want to change 'is' to in.

Answer 11. Thank you for spotting the typo. We replaced 'is' by 'in'.

Page 8 line 219-221: *“we observed NOTCH2 in meiotic hFGCs”*

Question 12. Figure 4A & 4B, please provide the intensity quantification analysis of CDH1 and CDH2 expression to support the statement in Lines 193-198.

Answer 12. As suggested, we have performed signal intensity quantification for CDH1 and CDH2 and presented that in our **novel Fig 4C**. In addition, as requested by Reviewer #1 (Question 10) we have also performed RNA FISH for CDH1 and CDH2 to identify which cells express them in ovarian cords and primordial follicles (**new Fig 4D**).

Page 8 line 216-219: *“Quantification analysis of the levels of immunofluorescence of CDH1 and CDH2 at the interface between (pre)granulosa cells and hFGCs revealed that the deposition of CDH1 increased during the transition from ovarian cord to PF, whereas CDH2 levels remained comparable (Fig 4C).”*

Question 13. Figure 4D, please quantify the expression of TJP1 between hFGCs and pre-granulosa cells at different ages.

Answer 13. Following this suggestion, we have performed signal intensity quantification for TJP1 and presented that in our **novel Fig 5A**.

Page 8 line 226-229: *“Regarding the formation of tight junctions (TJ) during the transition between ovarian cords and PFs, the expression of TJ protein TJP1 significantly increased between 16-18WPF at the contact points between the cell-membrane of hFGCs and (pre)granulosa cells and remained high in PFs at 20WPF (Fig 5A).”*

Question 14. Figure 4, did the authors notice any changes in organelle enrichment (mitochondrial, golgi, etc.) during the formation of tight junctions and cell-to-cell adhesion? To test this, the authors may want to analyze the number of Golgi/mitochondrial per cell in different aged human cortical.

Answer 14. We evaluated mitochondria enrichment using TOM20 antibody and enrichment of Golgi complex using GM130 antibody in different ages of ovaries, but we are unable to provide a meaningful quantification due to the fact that we are analyzing paraffin sections and this will not represent the whole cell. We do provide that IF data in **new Fig 2C**, when we present the data on the intercellular bridges. We have now also included the mitochondria localization in the Graphical Abstract.

Page 5 line 136-141: *“Next, we investigated the localization of mitochondria (marked by TOM20) and Golgi complex (marked by GM130) in hFGCs between 16-20WPF (Fig 2C). The (GM130+) Golgi complex did not display increased enrichment in specific hFGCs in ovarian cords and PFs, however (TOM20+) mitochondria showed increased expression and accumulation in hFGCs during the transition between ovarian cords and PFs (Fig 2C).”*

August 17, 2023

RE: Life Science Alliance Manuscript #LSA-2023-01926-TR

Prof. Susana M Chuva de Sousa Lopes
Leiden University Medical Center
Dept. of Anatomy and Embryology
Einthovenweg 20, 2300 ZC Leiden
Leiden 2300 RC Leiden
Netherlands

Dear Dr. Chuva de Sousa Lopes,

Thank you for submitting your revised manuscript entitled "Cell-cell interactions during formation of primordial follicles in humans". We would be happy to publish your paper in Life Science Alliance pending final revisions necessary to meet our formatting guidelines.

- please address the final Reviewer 1's comment
- please add ORCID ID for corresponding (and secondary corresponding) author--you should have received instructions on how to do so
- please consult our manuscript preparation guidelines <https://www.life-science-alliance.org/manuscript-prep> and make sure your manuscript sections are in the correct order
- please add your main, supplementary figure, and table legends to the main manuscript text after the references section
- please add a callout for Fig S1A,B ; Fig S2A-D to your main manuscript text;

A. FINAL FILES:

B. MANUSCRIPT ORGANIZATION AND FORMATTING:

Sincerely,

Reviewer #1 (Comments to the Authors (Required)):

The revised version of the manuscript by Czukiewska and colleagues is much improved and is a lot more rigorous regarding the conclusions and limitations of this study. The authors took into account all of my comments and I don't have any additional criticisms.

The formation of primordial follicles from multinucleated hFGCs is now clearly hypothetical (Figure 5F). It is still very surprising though. I am wondering whether the authors could speculate in the Discussion section, whether these multinucleated syncytia could be "nurse cells" remnants (even though they enter meiosis) rather than future PFs? Of course, it is just an optional suggestion.

Reviewer #2 (Comments to the Authors (Required)):

The authors have thoughtfully addressed all the questions and revised the manuscript accordingly. The reviewer has no more questions regarding the new version. Therefore, the revised manuscript is recommended for publishing in LSA.

Reviewer #1 (Comments to the Authors (Required)):

The revised version of the manuscript by Czukiewska and colleagues is much improved and is a lot more rigorous regarding the conclusions and limitations of this study. The authors took into account all of my comments and I don't have any additional criticisms.

We thank the Reviewer #1 for the positive remarks and constructive comments.

Question 1. The formation of primordial follicles from multinucleated hFGCs is now clearly hypothetical (Figure 5F). It is still very surprising though. I am wondering whether the authors could speculate in the Discussion section, whether these multinucleated syncytia could be "nurse cells" remnants (even though they enter meiosis) rather than future PFs? Of course, it is just an optional suggestion.

Answer 1. We fully agree with the suggestion of the Reviewer and we have now included this speculation in the Discussion section.

Page 9 line 267-270: "The formation of PFs from multinucleated hFGCs is hypothetical. It remains to be uncovered whether multinucleated hFGC-syncytia in human function in a similar way to nurse FGCs-oocyte described in mice (Niu & Spradling, 2022), even though they are capable of entering meiosis, rather than becoming future PFs."

Reviewer #2 (Comments to the Authors (Required)):

The authors have thoughtfully addressed all the questions and revised the manuscript accordingly. The reviewer has no more questions regarding the new version. Therefore, the revised manuscript is recommended for publishing in LSA.

We thank the Reviewer #2 for the positive remarks and constructive comments.

August 21, 2023

RE: Life Science Alliance Manuscript #LSA-2023-01926-TRR

Prof. Susana M Chuva de Sousa Lopes
Leiden University Medical Center
Dept. of Anatomy and Embryology
Einthovenweg 20, 2333 ZC Leiden
Leiden 2333 ZC Leiden
Netherlands

Dear Dr. Chuva de Sousa Lopes,

Thank you for submitting your Research Article entitled "Cell-cell interactions during formation of primordial follicles in humans". It is a pleasure to let you know that your manuscript is now accepted for publication in Life Science Alliance. Congratulations on this interesting work.

DISTRIBUTION OF MATERIALS:

Again, congratulations on a very nice paper. I hope you found the review process to be constructive and are pleased with how the manuscript was handled editorially. We look forward to future exciting submissions from your lab.

Sincerely,
